# Evaluation of the Efficacy of BBIBP-CorV Inactivated Vaccine Combined with BNT62b2 mRNA Booster Vaccine

**DOI:** 10.3390/diagnostics13030556

**Published:** 2023-02-02

**Authors:** Éva Rákóczi, Gusztáv Magócs, Sára Kovács, Béla Nagy, Gabriella Szűcs, Zoltán Szekanecz

**Affiliations:** 1Department of Rheumatology, Faculty of Medicine, University of Debrecen, 4023 Debrecen, Hungary; 2General Practitioner Service, 4283 Létavértes, Hungary; 3Department of Laboratory Medicine, Faculty of Medicine, University of Debrecen, 4023 Debrecen, Hungary

**Keywords:** COVID−19, BBIBP-CorV, BNT62b2, booster, vaccine

## Abstract

Objectives: In this prospective study, SARS-CoV−2 spike protein specific total immunoglobulin (Ig) levels were analyzed before and after BNT162 b2 mRNA booster vaccination in individuals previously administered with two doses of BBIBP-CorV vaccine in comparison to immunized participants with three doses of BNT162 b2 vaccination. Methods: Sixty-one Caucasian volunteers (39 females, 22 males) vaccinated by BBIBP-CorV were included (mean age: 63.9 years). Sixty-one patients (41 females, 20 males) as controls were vaccinated with BNT162b2 (mean age: 59.9 years). Both groups received the third booster BNT162b2 vaccine. Total anti-SARS-CoV−2 S1-RBD Ig levels were measured by an immunoassay (Roche Diagnostics) and their calculated ratios after/before booster dose were compared between the two groups. Results: At baseline, significantly lower anti-SARS-CoV−2 S1-RBD total antibody levels were determined after initial immunization by two doses of inactivated BBIBP-CorV compared to BNT62b2 mRNA vaccine (p < 0.001). After BNT162b2 boosters, similarly high total Ig levels were detected in both the heterologous (27,195 [15,604–42,754] BAU/mL, *p* < 0.001) and the homologous booster cohort (24,492 [13,779−42,671] BAU/mL, *p* < 0.001) compared to baseline. Hence, the ratio of after/before total Ig levels was significantly higher with heterologous vs homologous immunization (*p* < 0.001). Conclusion: To address the concept that basic BBIBP-CorV vaccination is not as effective as BNT162b, we analyzed the effect of heterologous vaccination with BNT162b2. Our results suggest that BNT162b2 can successfully boost the effects of two-dose BBIBP-CorV vaccination.

## 1. Introduction

A total of 608 million people have been infected with the SARS-CoV−2 virus worldwide as of August 2022, and over 6.4 million patients have died of this infection [1]. Vaccination is the best method to protect against that virus since specific effective antiviral therapy has yet to be developed. Hungary has established a risk-based vaccination program for the COVID−19 vaccines when one type of vaccine, i.e., BNT162b2, was not available for all individuals [2]. The principle of four priority target groups had been established regarding the urgent need for vaccination: (1) healthcare and social care providers; (2) individuals with COVID−19 associated risks (e.g., over 60 years of age, or age between 18–59 years but with underlying disease or comorbidities); (3) those working in critical infrastructure (e.g., kindergarten and school teachers, trade workers); and (4) those 18–59 years of age who are not included in any of these groups. In Hungary, the first available anti-SARS-CoV−2 vaccine was the BNT162b2 mRNA vaccine, thus all healthcare professionals and social workers were immunized by this vaccine. For the second group, the BBIBP-CorV inactivated vaccine was available at the largest quantity and was based on the Hungarian Public Health Centre (NPH) guide, and general practitioners applied this immunization for this population. According to the original study report, BBIBP-CorV was effective in individuals between 18–59 years. Because of the non-availability of mRNA and adenoviral anti-SARS-CoV−2 vaccines in Hungary at the beginning of the vaccination program, several individuals above the age of 60 had to be vaccinated using the BBIBP-CorV vaccine [2]. At that time there were no available data on the efficacy of the BBIBP-CorV vaccine in the elderly [3]. In some later studies, a diminished response to BBIBP-CorV vaccine above the age of 65 was reported [4]. Therefore, in Hungary, similarly to other countries, where BBIBP-CorV was applied to elderly citizens, the possibility of the use of a heterologous booster was highly awaited.

Since then, the third booster dose of vaccination was also recommended and introduced during the third wave of the COVID−19 pandemic in 2021. As in some studies, BNT62b2 showed a superior quantitative efficacy over BBIBP-CorV [5], and it needed to be further determined whether a homologous or a heterologous regimen using the BNT62b2 booster for those who previously received two BBIBP-CorV or BNT62b2 doses, respectively, should be applied to raise anti-SARS-CoV−2 S1-RBD antibody titers for improving humoral immune response. To date, there is only a limited amount of clinical data regarding the efficacy of the BBIBP-CorV inactivated vaccine combined with the BNT62b2 (Pfizer-BioNTech) mRNA vaccine as a booster vaccine in non-Caucasian populations [5,6,7,8,9,10,11]. In heterologous protocols, BNT62b2 as a booster was applied after a basic regimen using BBIBP-CorV with a temporal separation of several months. There was a robust increase in anti-spike antibody production after receiving the heterologous BNT162b2 booster. Using the heterologous regimen, anti-spike humoral immunity, in most studies, was significantly stronger than in individuals receiving either no booster or a homologous booster with another dose of BBIBP-CorV. In addition, in these non-Caucasian populations, heterologous booster regimens were safe and well-tolerated [6,8,9,10,11].

Our aim was to conduct a real-life study in which a typical European population was analyzed after two different types of vaccinations. Hence, we sought to investigate anti-SARS-CoV−2 total immunoglobulin (Ig) concentrations after two doses of BBIBP-CorV and BNT62b2 booster vaccines in a European cohort. We wished to compare humoral immunity after this heterologous regimen to the homologous regimen of three BNT62b2 vaccinations. We also wanted to study the efficacy and safety of these different vaccination schedules in individuals belonging to different age groups. To the best of our knowledge there have only been a very few studies carried out in Caucasian populations where the BBIBP-CorV vaccine was included in heterologous vaccination regimens [12].

## 2. Subjects, Methods, and Study Design

### 2.1. Study Design

In this prospective study, we examined the total immunoglobulin (Ig) levels against SARS-CoV−2 spike protein 1 receptor-binding domain (anti-SARS-CoV−2 S1-RBD) after the third (second booster) dose of the BNT162b2 vaccine in the BBIBP-CorV vaccinated group (heterologous regimen) compared to a control group of subjects receiving three doses of BNT162b2 (homologous regimen). Following the initial immunization period, a survey was conducted to monitor vaccine efficacy via the incidence of infection and hospitalization by patient questionnaire. Booster immunization was carried out at least 4 months after the first two doses of basic immunization, and baseline (pre-booster) antibody levels were measured on the same day of the administration of booster immunization (first sampling). Following the booster dose, another serum sample was collected after 30 days to determine the induced total level of anti-SARS-CoV−2 spike protein antibodies (second sampling). Figure 1 shows the study design.

Exclusion criteria included age < 18 years, known primary immunodeficiency, malignancy, and ongoing immunosuppressive therapy. As part of the regular clinical practice, all participants received the third vaccine dose at a general practitioner’s (GP) office. We double-checked the history of these participants and none of these individuals had received any other (anti-SARS-CoV−2) vaccinations prior to this study.

We also thoroughly followed the adverse effect of vaccinations and the clinical status of these subjects via regular personal check-ups. We monitored for SARS-CoV−2 infection by using virus-specific PCR tests in case of any suspicious symptoms to exclude any ongoing infection during the course of the study. Eventually, only one patient (1.67%) was infected with the SARS-CoV−2 virus, showing mild symptoms in the BBIBP-CorV vaccinated group but without requiring any hospitalization.

We did not use a sample size calculator for this study. Instead, we randomly selected individuals into this study and recruited patients until we could create age- and sex-matched sub-cohorts to compare anti-SARS-CoV−2 Ig levels after heterogenous and homologous vaccination.

Our study included a total of 122 participants; 61 patients served as the study group and received basic vaccination with BBIBP-CorV, and a control group of 61 patients were vaccinated by two doses of BNT162b2. The demographic data of the patients, including comorbidities, are shown in Table 1. The two groups were age-matched as participants in the heterologous-vaccinated cohort, as the study group had a mean age of 63.9 ± 12.61 years, while subjects in the control, homologous-vaccinated group showed a mean age of 59.9 ± 12.92 years (*p* = 0.090). There were 39 female patients (62.9%) in the study group and 41 (67.2%) in the control group (*p* = 0.849). The most common co-morbidity was diabetes mellitus in both groups (21.3% vs. 16.4%, *p* = 0.643). In addition, co-morbidities of patients including cardiovascular disease (coronary heart disease, hypertension), autoimmune diseases (chronic thyreoditis, Graves-Basedow disease, ulcerative colitis, Crohn’s disease), chronic respiratory diseases (asthma bronchiale, chronic obstructive pulmonary disease), chronic renal insufficiency and diabetes mellitus did not differ significantly between the study cohorts. The patients had no history of COVID−19: 61 (39 females and 22 males) of them were immunized with the inactivated BBIBP-CorV vaccine (Sinopharm) and the 61 control subjects (41 females and 20 males) received two doses of BNT162b2 mRNA vaccine (Pfizer-BioNTech).

In both groups, the first and second doses were administered 21 days apart. The additional BNT162b2 mRNA booster vaccine was administered at least four months (132–309 days) after the basic immunization according to the NPH guide. For further analysis, participants were sub-grouped based on their age into two sub-cohorts: ≤60 years and >60 years of age, and a correlation test between anti-SARS-CoV−2 S1-RBD antibody levels and time intervals between the second and the booster immunization was performed.

The study was approved by the Scientific and Research Ethics Committee of the University of Debrecen, the Ministry of Human Capacities (32568–7/2020/EÜIG), and the Hungarian Medical Research Council (ETT-TUKEB; IV/4050–3/2022/EKU). Informed consent was obtained from all participants. The study was performed according to the Declaration of Helsinki.

### 2.2. Laboratory Methods

An automated Cobas^®^ Anti-SARS-CoV−2 S serology test (Roche Diagnostics, Mannheim, Germany) was used for measuring total Ig levels in the serum samples in order to determine SARS-CoV−2 S1-RBD-specific antibody titers before and after booster vaccination. The test was performed according to the manufacturer’s instructions. Seropositivity was considered according to the official cut-off value of the test (≥0.8 BAU/mL).

### 2.3. Statistical Analysis

A Kolmogorov–Smirnov test was used for the evaluation of the normality of the data. Results were expressed as mean ± standard deviation (SD) or median with interquartile range (IQR), as appropriate. To compare the data of two groups, a Wilcoxon test was used for serology results and a chi-squared test was applied for demographic parameters. The relationship between the variables was examined using a Spearman’s correlation. The *p* < 0.05 probability value was accepted as significant. The „R project” mathematical software (RStudio, Boston, MA, USA) [13] was used for the statistical analysis.

## 3. Results

### 3.1. Antibody Responses after Different Booster Vaccination Regimens

At baseline, anti-SARS-CoV−2 S1-RBD antibody levels were significantly (*p* < 0.0001) lower in the BBIBP-CorV compared to the BNT162b2 groups (20.8 [6.6–98.7] BAU/mL vs 903,4 [528.8–1811.7] BAU/mL, respectively) (Figure 2). Both groups demonstrated significantly higher antibody levels after booster vaccination (BBIPB-CorV cohort: 27195 [15,604–42,754] BAU/mL; BNT162b2 controls: 24,492 [13,779, 42,671] BAU/mL, respectively), compared to their own baseline (*p* < 0.0001) (Figure 2). When we further compared the increasing tendency in titers between the two groups, significantly higher ratios of after/before total antibodies were found in the heterologous vs homologous cohorts (*p* < 0.0001) (Figure 3). Based on these data, we found that the BNT162b2 booster vaccine can effectively augment the level of anti-SARS-CoV−2 S1-RBD antibody levels even after a relatively low Ig level induced by two initial BBIPB-CorV doses.

### 3.2. Comparison of Anti-SARS-CoV−2 Total Ig Levels between Age Groups

Next, we wanted to investigate whether there was any difference between those who were under or over 60 years of age regarding induced humoral immune response before and after booster vaccination. Surprisingly; age did not affect anti-SARS-CoV−2 total Ig levels in these cohorts: (1) there was no difference between pre-booster levels in those who were older than 60 years compared to younger (≤60 years) study participants within either study group; and (2) post-booster total Ig levels did not elevate at a higher extent in younger subjects, especially in the homologous vaccination group (Figure 4). Accordingly, only the type of vaccine (but not age) influenced the baseline anti-SARS-CoV−2 total Ig levels, causing lower titers in BBIBP-CorV group, which could be improved by the BNT162b2 booster vaccine.

### 3.3. Correlation between Antibody Response and the Timing of Booster Immunization

The time interval between the second and third vaccination ranged from 132–309 days. When we analyzed the relationship between anti-SARS-CoV−2 total Ig levels after booster vaccination and the time period that had lapsed between the second and third dose in both groups, a moderate but significant correlation was observed between these variables (Spearman’s R = 0.22; *p* = 0.015), suggesting that the efficacy of the booster vaccine was the most effective by 6–8 months between basic and booster vaccinations; in the BBIBP-CorV group this took about 8 months longer (Figure 5). Based on these findings, the timing of the administration of the booster vaccine may also have had an impact on induced anti-SARS-CoV−2 total Ig levels that can ensure the sufficient humoral response after immunization.

### 3.4. Clinical Course of Vaccinations

The three doses of vaccines were successfully administered to all subjects and were found to be safe and well tolerated. During the basic immunization regimen, only one patient (1.7%) acquired SARS-CoV−2 infection with mild symptoms in the BBIBP-CorV group 3 days after receiving the second vaccine dose. Importantly, this patient did not require any hospitalization. No COVID−19 developed in the control group or in any of the two groups after the third dose. In addition, we did not observe any serious or clinically significant side effects.

## 4. Discussion

In this real-world study, we compared the immunogenicity of the third BNT62b2 booster vaccination in individuals who had previously received either two BBIBP-CorV vaccinations (heterologous regimen) or two BNT62b2 vaccine disease (homologous regimen) vaccinations. We found that both regimens resulted in high total post-vaccination anti-SARS-CoV Ig levels compared to baseline. Yet, the ratio of after/before total Ig levels was significantly higher applying the heterologous compared to the homologous immunization regimen. Moreover, age did not influence anti-SARS-CoV−2 total Ig levels in either of the two regimens, as there was no difference between pre-booster or post-booster levels in those older than 60 years compared to younger study participants. Finally, the efficacy of the third booster vaccine was the most effective 6–8 months after the basic schedule and there was a correlation between the timing of the administration of booster vaccine and anti-SARS-CoV−2 total Ig levels. Thus, a third dose of BNT162b2 can successfully boost the effects of two-dose BBIBP-CorV vaccination. The three doses of vaccines were safe and well tolerated.

During the COVID−19 pandemic, the importance of clinical laboratory tests has emerged to evaluate the humoral response following different types of vaccines [14], but they estimated the incidence of SARS-CoV−2 infection in patients with malignancy and under anti-cancer therapy as well [15]. Numerous clinical studies have compared the serological status of variable populations among distinct clinical conditions, such as after receiving primary vaccination with BNT162b2 and a homologous booster with or without previous SARS-CoV−2 infection [14,15,16,17,18,19], or following other vaccines (e.g., Sputnik V) [20,21]. Salvagno et al. [17; 18] suggested that a significant difference existed in post-mRNA COVID−19 vaccine immune responses in anti-SARS-CoV−2 seronegative versus seropositive subjects at baseline. This seemed to be dependent upon age and sex in seronegative subjects, as well as of baseline anti-SARS-CoV−2 antibody levels in seropositive patients [17,18]. Urlaub et al. [19] reported that individuals who got infected as early as 10 days after their first immunization show antibody levels comparable to fully vaccinated individuals. Subsequently, the immunogenicity of various heterologous immunizations was assessed [12]. Pascuale et al. [12] assessed the immunogenicity and reactogenicity of 15 vaccine combinations in more than 1300 participants. They evaluated anti-spike IgG response and virus neutralizing titres and observed that a number of heterologous vaccine combinations were equivalent or superior to homologous regimens. The highest antibody response was induced by mRNA vaccines as the second dose. The authors concluded that the use of different vaccine combinations would achieve wide vaccine coverage in the shortest possible time. However, there have been only very few trials in this respect on the BBIPB-CorV, as this vaccine has been used in only a limited number of countries [6,7,22,23,24,25], including Hungary [21,26,27,28]. The original phase I/II trial confirmed the efficacy of BBIPB-CoV in 18–59-year-old subjects [3], which was further supported by other studies [12,22,23,24,25]. However, diminished efficacy has been suggested in elderly individuals (>60 years) [4]. In March of 2022, the World Health Organization (WHO) updated interim recommendations for the use of BBIBP-CorV [29]. According to the recommendation, no published data on the efficacy of this immunization was available among elderly (>60 years of age) patients [29]. We and others have recently confirmed that in comparison to the BNT162b2, mRNA−1273 and Ad26.COV2.S vaccines, BBIBP-CoV had lower efficacy and faster waning of effectiveness [27,28,30,31]. When comparing various strategies with different vaccines, Blanco et al. [31] found that BBIBP-CorV used as a second dose exhibited a significantly lower neutralizing response compared to other protocols. These authors also suggested that BBIBP-CorV should not be recommended as a second dose. Since in Hungary in 2020–2021 the BBIBP-CorV vaccine was widely used for basic immunization in individuals over the age of 60 and third vaccination doses were introduced to the nation, we wished to assess whether a second booster using a different (BNT162b2) vaccine would increase the immunogenicity of BBIBP-CorV to the level obtained by the three-dose homologous protocol using BNT162b2 only.

Very few heterologous vaccination studies using BNT162b2 after a basic regimen of two BBIBP-CorV vaccinations have been performed in non-Caucasian populations [6,7,8,9,10,11]. For example, Kanokudom et al. [11] applied a protein subunit vaccine after two-dose basic vaccination regimens that included BBIBP-CorV. The study of Hueda-Zavaleta et al. [8] carried out in Peru applied a temporal separation of 7 months between BBIBP-CorV and BNT162b2 to healthcare workers. There was a robust increase in humoral immunity after receiving the heterologous BNT162b2 booster that was more pronounced in individuals not previously infected by SARS-CoV−2 [8]. In another study conducted in Peru, Vargas-Herrera et al. [10] found that the geometric means IgG levels increased significantly after boosting with BNT162b2 after BBIBP-CorV. Heterologous immunization resulted in a 17-fold increase in anti-spike protein antibodies. Two doses of BBIBP-CorV boosted with BNT162b2 produced a stronger IgG antibody response than the homologous BNT162b2 regimen [10]. Moghnieh et al. [6] carried out a similar study in Lebanon. A heterologous booster vaccination was significantly associated with higher anti-spike IgG geometric mean titers compared to that after homologous BNT162b2 immunization in COVID−19-naïve subjects. These authors also concluded that in individuals who have already received BBIBP-CorV, mixing BBIBP-CorV and BNT162b2 might overcome the low immunogenicity induced by BBIBP-CorV alone. This may be especially useful for the protection against emerging variants [6]. In the study of Park et al. [9] that included Korean health professionals, the BNT162b2 booster after the second dose of BBIBP-CorV, compared to no booster or to a third BBIBP-CorV booster, was associated with more effective protection against laboratory-confirmed COVID−19. Importantly, heterologous vaccination with BNT162b2 after BBIBP-CorV was safe and well-tolerated in all of these studies [6,7,8,9,10,11].

However, to the best of our knowledge, there has been only a very few studies on heterologous booster vaccination following a basic regimen using BBIBP-CorV in a Caucasian population. The study of Pascuale et al. [12] described above also included BBIPB-CoV in the assessed regimens.

The three doses of vaccines were successfully administered to all subjects and were found to be safe and well tolerated, as reported by others [6,8,9,10,11,12,17,18,19,27,30,32,33,34]. Before the booster, the baseline anti-S1-RBD levels were higher in the BNT162b2 group compared to the BBIBP-CorV group, suggesting that, indeed, the immunogenicity of BBIBP-CorV may be diminished compared to BNT162b2. As described above, this difference has been suggested previously [27,30]. The BNT162b2 booster significantly increased anti-SARS-CoV−2 production in both groups to the same level. Thus, BNT162b2 may effectively boost humoral immunogenicity in subjects previously receiving a BBIBP-CorV basic protocol to the extent obtained by three BNT162b2 shots. Furthermore, when we studied the effect of age on these serology test results, there were lower pre-booster levels in those who were older than 60 years of age compared to younger study participants in either group, while no larger elevation in total Ig levels was observed in younger individuals, especially in the homologous vaccination group. These data suggest that in general, BBIBP-CorV is not always sufficient in the entire population to trigger humoral response; however, the BNT162b2 booster vaccine can augment anti-SARS-CoV−2 antibody levels to provide immunization against SARS-CoV−2 infection even in older subjects. In contrast to our findings, a previous study concluded that anti-SARS-CoV−2 antibody production after Sinopharm vaccination was decreased with increasing age [27,30]. Nevertheless, there is increased evidence that vaccination boosters help to increase antibody levels in the high-risk groups, such as older people and those with comorbidities, supporting the value of COVID−19 booster vaccination [27,30,32,33,34].

We found that the BNT162b2 booster led to a significant increase in spike protein antibodies. The ratio of antibody levels for both groups of vaccines showed the same increase, confirming the ability to boost BBIBP-CorV antibodies for the BNT162b2 vaccine. However, only a modest but significant relationship was seen between antibody response and the length of time interval between the second and third dose, thus the timing of the administration of booster vaccine may also have an impact on induced anti-SARS-CoV−2 total Ig levels that can ensure the sufficient humoral response after immunization.

Our study has certain strengths and limitations. The major strength of this study is that, to the best of our knowledge, this is one of the very first studies on the use of a BNT162b2 heterologous booster in individuals who had received two BBIBP-CorV vaccinations in a Caucasian population. We also consider it to be a strength that we compared this heterologous booster regimen with the homologous protocol of three shots of BNT162b2. We also compared the efficacy and safety of this vaccination regimen in various age groups. Limitations might include the single-centre nature of the study. Furthermore, we only measured anti-SARS-CoV S1 RBD concentration as a biomarker of humoral immunity against the vaccine.

## 5. Conclusions

In conclusion, we confirmed that the basic immunization regimen of two doses of BBIBP-CorV is less powerful in inducing anti-SARS-CoV−2 RBD total Ig antibody levels than two doses of BNT162b2. We wished to determine whether a third dose of BNT162b2 after two doses of BBIBP-CorV would effectively booster total Ig levels. Indeed, in a heterologous regimen, BNT162b2 effectively boosted anti-SARS-CoV−2 RBD humoral immune responses compared to the homologous regimen of three BNT162b2 doses. The humoral responses were age-independent but correlated with the timing of vaccine administration. The booster vaccination was safe and tolerable. Our results suggest that BNT162b2 can successfully boost the effects of two-dose BBIBP-CorV vaccination. In order to increase the knowledge on booster vaccinations for ensuring accurate, responsible clinical decisions, further studies with even larger numbers of participants are required in the near future. More studies including Caucasian individuals receiving a BBIBP-CorV vaccination schedule should also be conducted.

## Figures and Tables

**Figure 1 diagnostics-13-00556-f001:**
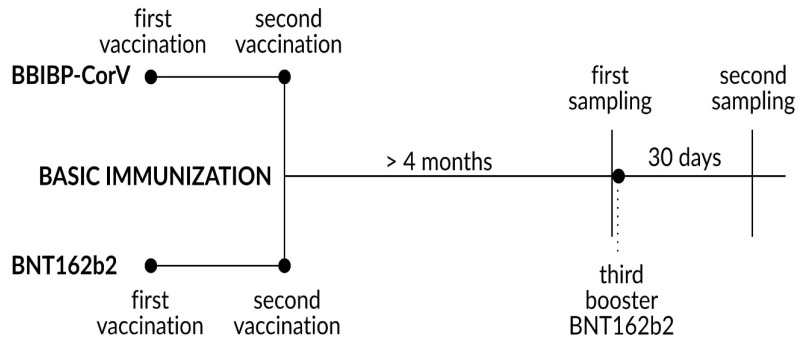
Schematic figure of the study design.

**Figure 2 diagnostics-13-00556-f002:**
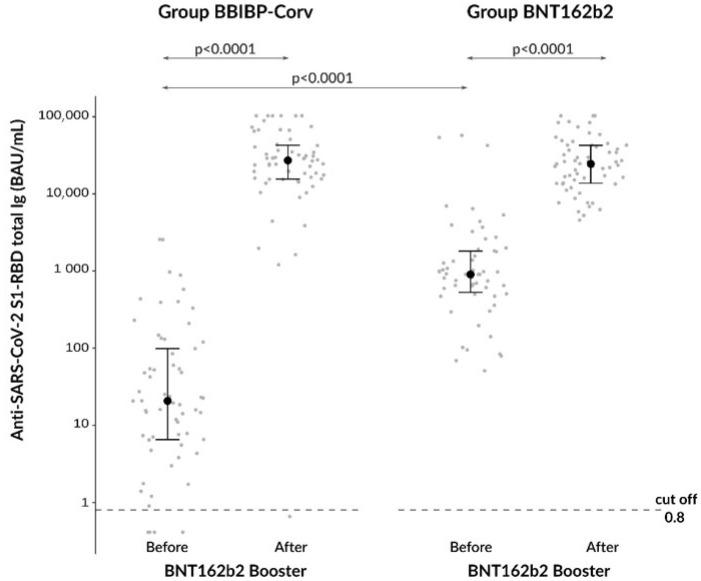
Comparison of anti-SARS-CoV−2 total Ig levels before and after booster vaccination between the BBIBP-CorV and BNT162b2 groups.

**Figure 3 diagnostics-13-00556-f003:**
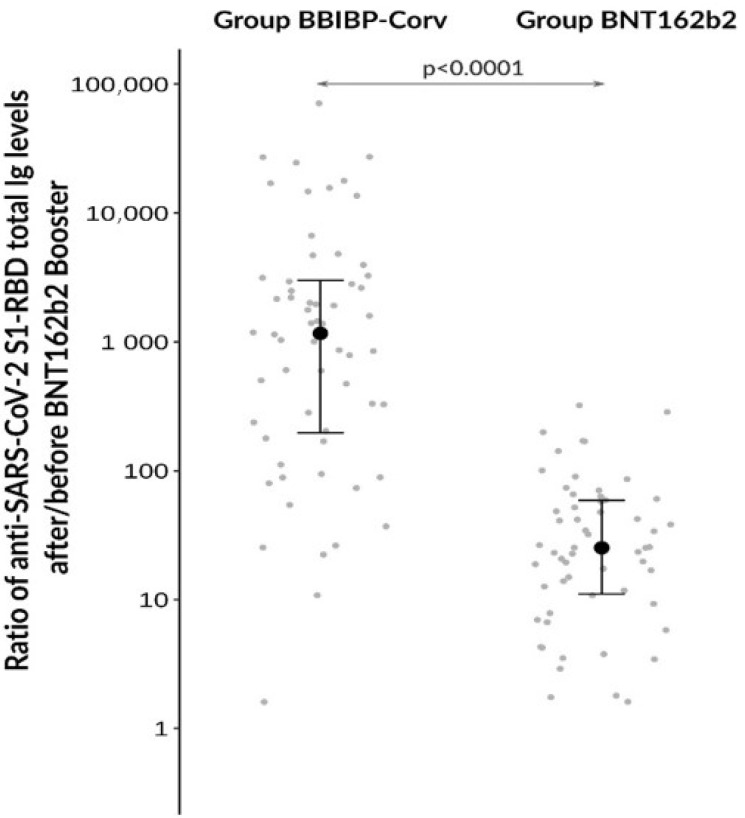
Analysis of the ratio of anti-SARS-CoV−2 S1-RBD total Ig levels after/before BNT162b2 booster vaccine between BBIBP-CorV and BNT162b2 groups.

**Figure 4 diagnostics-13-00556-f004:**
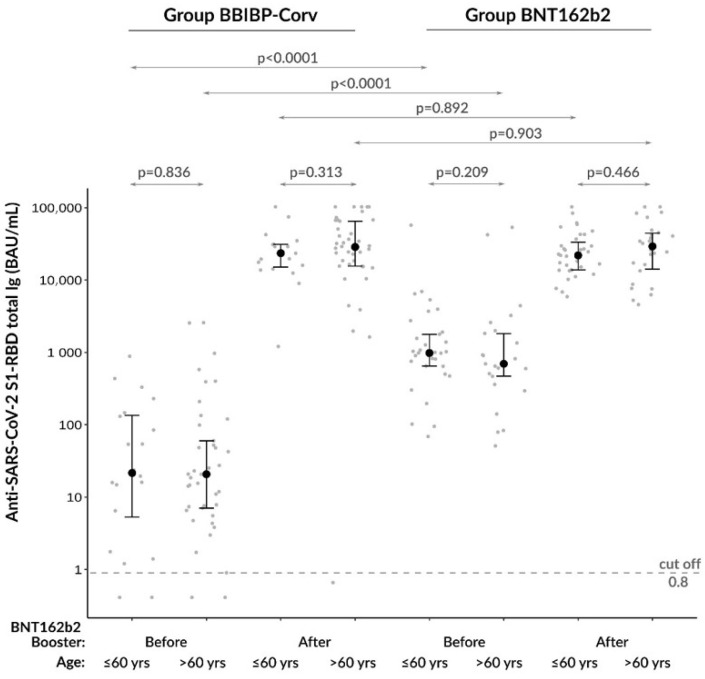
Investigation of the effect of age on anti-SARS-CoV−2 S1-RBD total Ig titers within both study groups before and after booster vaccination.

**Figure 5 diagnostics-13-00556-f005:**
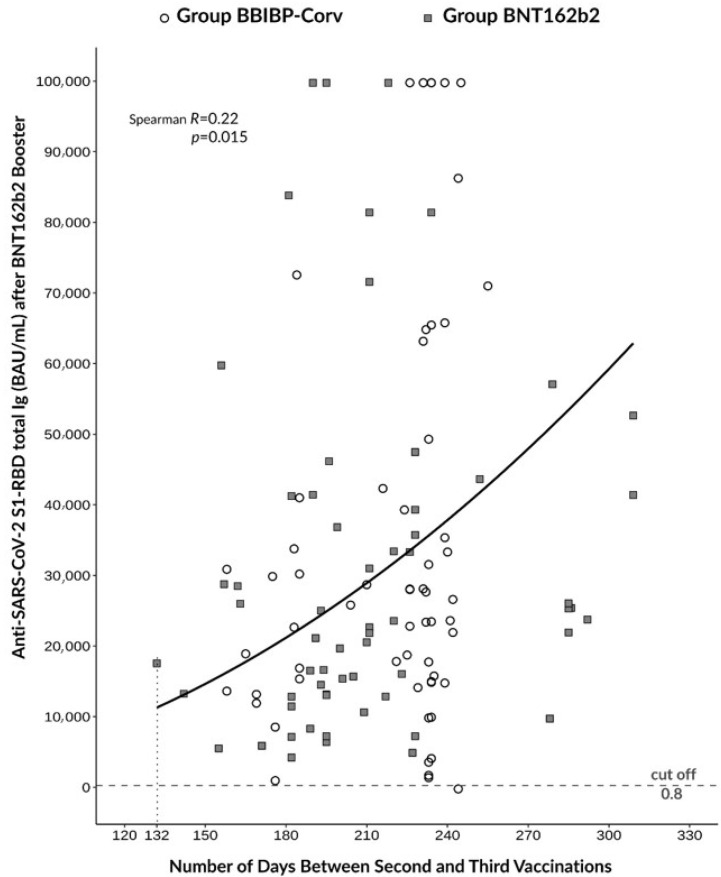
Correlation between antibody response and the length of time interval between second and third dose of vaccines.

**Table 1 diagnostics-13-00556-t001:** Demographic data of the patients.

	All Patients	BBIBP-CorV Group	BNT162b2 Group	*p*
All patients (n)	122	61	61	
Age [year] [mean (SD)]	61.9 (12.87)	63.9 (12.61)	59.9 (12.92)	0.090
Gender [female (%)]	80 (65.6)	39 (62.9)	41 (67.2)	0.849
All comorbidities [n (%)]	90 (73.8)	45 (73.8)	45 (73.8)	~1.0
Cardiovascular [n (%)]	86 (70.5)	43 (70.5)	43 (70.5)	~1.0
Autoimmune [n (%)]	18 (14.8)	9 (14.8)	9 (14.8)	~1.0
Respiratory [n (%)]	13 (10.7)	7 (11.5)	6 (9.8)	~1.0
Renal [n (%)]	6 (4.9)	3 (4.9)	3 (4.9)	~1.0
Diabetes mellitus [n (%)]	23 (18.9)	13 (21.3)	10 (16.4)	0.643

## Data Availability

Data is available upon request.

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
