# Peer review of "Evaluation of the Efficacy of BBIBP-CorV Inactivated Vaccine Combined with BNT62b2 mRNA Booster Vaccine"

_diagnostics, 2023, doi:10.3390/diagnostics13030556_

Round 1

Reviewer 1 Report

Thank you for undertaking such an important study. Although this study could have a critical contribution to its field, I should advise a major revision because of some fundamental errors in the methodology. Therefore, please address the following comments:

1) Regarding the study sample size, the authors mentioned that they did not use a sample size calculator. Thus, the 122 participants included in this study are insufficient to address the study assumption.

2) The authors did not measure an essential component of immunity against COVID-19, the neutralizing antibodies. 

3) Did the authors use any database to check a possible previous vaccine shot for the participants?

4) What parameters did you follow up on and assess in the 4 months between the second and third doses?

5) Did you record any comorbidities for your participants? This is a matter of high importance, as some of the underlying comorbidities affect the antibody response of individuals.

6) As already fourth and fifth doses of COVID-19 vaccines are suggested by the health authorities worldwide, did you gather any data on these additional booster doses?

Author Response

Reviewer 1

Thank you for undertaking such an important study. Although this study could have a critical contribution to its field, I should advise a major revision because of some fundamental errors in the methodology. Therefore, please address the following comments:

We are grateful for the Reviewer to critically revised our manuscript. Here we would like to respond to the questions and comments one-by-one. We hope that our answers will be acceptable in the light of our replies.

1) Regarding the study sample size, the authors mentioned that they did not use a sample size calculator. Thus, the 122 participants included in this study are insufficient to address the study assumption.

R1: In this study, we recruited consecutive healthy patients who attended a Hungarian GP office to require the vaccinations. Thus, our aim was to conduct a “real life” study in which a typical European population was analyzed after two different types of vaccinations. This information is now included in the manuscript (page 5). Only those patients were excluded who was not COVID-19 naïve. This latter condition was not easy to set as many subjects had already been infected by the SARS-CoV-2 and we sought to investigate only the effect of homologous and heterologous vaccines via anti-SARS-CoV-2 total Ig levels.

2) The authors did not measure an essential component of immunity against COVID-19, the neutralizing antibodies. 

R2: We agree with the comment that we did not measure the amount of neutralizing antibodies but based on the official description of this test (Roche Diagnostics, Mannheim, Germany), the BAU/mL values of anti-SARS-CoV-2 total Igs reflect to the level of neutralizing Igs in sera. That is why we did not find important to directly analyze such antibodies in these samples.  

3) Did the authors use any database to check a possible previous vaccine shot for the participants?

R3: We have double-checked the history of these participants and none of these individuals had received any other (anti-SARS-CoV-2) vaccinations prior to this study. This is now included in the text (page 6, in red).

4) What parameters did you follow up on and assess in the 4 months between the second and third doses?

R4: We thoroughly followed the adverse effect of vaccinations and the clinical status of these subjects if they had been infected by SARS-CoV-2 despite the administration of vaccination(s) via regular personal check-ups and using SARS-CoV-2 specific PCR tests in case of any suspicious symptoms to exclude any ongoing infection during the study course. Eventually, only one patient (1.67%) was infected by the SARS-CoV-2 virus showing mild symptoms in the BBIBP-CorV vaccinated group without any hospitalization. This information is now added to the text (page 6)              

5) Did you record any comorbidities for your participants? This is a matter of high importance, as some of the underlying comorbidities affect the antibody response of individuals.

R5: Thank you for your helpful suggestion and accordingly, we have collected the comorbidities of these subjects and inserted these data into a new Table 1. There are no significant differences between the two groups according to any of the demographic variables (Table 1). Co-morbidities including cardiovascular disease (coronary heart disease, hypertension), autoimmune diseases (chronic thyreoditis, Graves-Basedow disease, ulcerative colitis, Crohn disease), chronic respiratory diseases (asthma bronchiale, chronic obstructive pulmonary disease), chronic renal insufficiency and diabetes mellitus did not differ significantly in BBIBP-CorV and control group. We also included this information in the text, in the Patients and methods section.

6) As already fourth and fifth doses of COVID-19 vaccines are suggested by the health authorities worldwide, did you gather any data on these additional booster doses?

R6: In Hungary, several people have only max. 3-4 vaccinations by now depending on if they have any subsequent comorbidities. However, this population have become quite diverse in that way as they have already received different other vaccines (such as viral vector-based vaccines) as a booster, thus these groups of subjects cannot be compared systemically with each other anymore. This present study was conducted when most Hungarian inhabitants have had only 2 shots.

Reviewer 2 Report

Dear Authors

The study by Éva Rákóczi and colleagues entitled " Evaluation of the efficacy of BBIBP-CorV inactivated vaccine  combined with BNT62b2 mRNA booster vaccine " investigated  SARS-CoV-2 spike protein-specific total immunoglobulin (Ig) levels, before and after BNT162b2 mRNA booster vaccination in individuals previously administered with two doses of BBIBP-CorV vaccine in comparison to immunized participants with three doses of BNT162b2 vaccination. They demonstrated significantly induced anti-SARS-CoV-2 S1-RBD antibody levels in BBIBP-CorV immunized subjects after booster BNT162b2 vaccination regardless of age similar to those with three BNT162b2 vaccines.

Overall, the text was well written. However, the content of this study is relatively simple, and the research is relatively superficial. The data of this study is the preliminary data of many other studies in the field of COVID-19, as the authors also mentioned. In my point of view, the study is not innovative enough to be published in diagnostics.

Major comments:

1.      The large volume of the abstract should be reduced for better understanding.

2.      As the authors mentioned in the text of the manuscript, this study is similar to many other studies which were done in many populations (references: 6, 8, 9, 10, 11, 12). So, there is no novelty in this content.

3.       I think the titer of the antibody should be measured before including the subjects in the study, as Ab titer in some of the subjects could be in saturation and high titer which the booster couldn't increase it.

4.      Why did the authors use Spearman's correlation test? Were all the variables nonparametric?

5.      The paragraph in Lines 306-316 was not related to this study. The authors should discuss more the study which was done by them, not the findings of another study.

Minor comments:

1.      In line 86: "in" was written twice. One of them should be removed.

2.      In line 142: "Informed consent was obtained from all participants" was mentioned before. It should be removed.

3.      In figure 2 legend: "anti-SARS-Cov-2" should be replaced by "anti-SARS-CoV-2 S1-RBD".

4.      In line 310: anti-spike protein antibodies is correct.

Author Response

Reviewer 2

Overall, the text was well written. However, the content of this study is relatively simple, and the research is relatively superficial. The data of this study is the preliminary data of many other studies in the field of COVID-19, as the authors also mentioned. In my point of view, the study is not innovative enough to be published in Diagnostics.

We are grateful for the Reviewer to critically revised our manuscript. Here we would like to respond to the questions and comments one-by-one. We hope that our answers will be acceptable in the light of our replies.

  1. The large volume of the abstract should be reduced for better understanding.

R1: We thank you for your helpful comment, and we now significantly shortened the abstract.

  1. As the authors mentioned in the text of the manuscript, this study is similar to many other studies which were done in many populations (references: 6, 8, 9, 10, 11, 12). So, there is no novelty in this content.

R2: We agree with the comment that our current study shows similarities with recent other publications, but these data have been produced in a European population which may have other genetic background, that is why we sought to conduct this study measuring anti-SARS-CoV-2 total Igs in response to two different types of booster vaccinations.

  1. I think the titer of the antibody should be measured before including the subjects in the study, as Ab titer in some of the subjects could be in saturation and high titer which the booster couldn't increase it.

R3: Our aim was to investigate the direct effect of vaccinations against SARS-CoV-2 infection, and it took some time for the vaccines to induce antibodies against the virus that is why we set this study as we described. We agree with the fact that in many cases the booster shot could not further increase the titer as the first two doses were quite effective and the level of Igs did not decrease as rapidly as it was expected.

  1. Why did the authors use Spearman's correlation test? Were all the variables nonparametric?

R4: Yes, indeed, all analyzed data were non-parametric based on normality test.

  1. The paragraph in Lines 306-316 was not related to this study. The authors should discuss more the study which was done by them, not the findings of another study.

R5: We fully agree, we now no longer discuss this Brazilian study. However, we think we already discussed our own study very extensively.

Minor comments:

  1. In line 86: "in" was written twice. One of them should be removed.

R1: Thank you, we corrected it.

  1. In line 142: "Informed consent was obtained from all participants" was mentioned before. It should be removed.

R2: Thank you, we removed one of the informed consent statements.

  1. In figure 2 legend: "anti-SARS-Cov-2" should be replaced by "anti-SARS-CoV-2 S1-RBD".

R3: Thank you for your advice, we have corrected in the article.

  1. In line 310: anti-spike protein antibodies is correct.

R4: Thank you for your advice, we have corrected in the article.

Reviewer 3 Report

Until now there are very few works in this area.The work was very well conducted . The proposal is very relevant because as we know at the beginning of vaccination we didn't have enough vaccines of the same brand so we had to receive vaccines with different formulations. There are countries where an individual has been immunized with up to three differents types of vaccines in three or four doses and even between the intervals of vacination  they were affected by a new variants. These studies are still in the beginning, but they must be conducted. Not only on the protection aspect but also on the immune response, not only related to humoral but also to cellular immunity. The group's studies must follow or others will emerge regarding the functionality of the antibodies produced. The relation of isotypes of immunoglobulins as IgG1, 2, 3 or 4 in relation to the individuals affected with the disease and after vaccination.Certainly the level and functionality of the isotypes will be different and there will probably be a difference between infection and vaccination with SARS-CoV-2 vaccines. More studies can be explored by the group or in partnership in the future.

Author Response

Reviewer 3

Until now there are very few works in this area.The work was very well conducted . The proposal is very relevant because as we know at the beginning of vaccination we didn't have enough vaccines of the same brand so we had to receive vaccines with different formulations. There are countries where an individual has been immunized with up to three differents types of vaccines in three or four doses and even between the intervals of vacination  they were affected by a new variants. These studies are still in the beginning, but they must be conducted. Not only on the protection aspect but also on the immune response, not only related to humoral but also to cellular immunity. The group's studies must follow or others will emerge regarding the functionality of the antibodies produced. The relation of isotypes of immunoglobulins as IgG1, 2, 3 or 4 in relation to the individuals affected with the disease and after vaccination.Certainly the level and functionality of the isotypes will be different and there will probably be a difference between infection and vaccination with SARS-CoV-2 vaccines. More studies can be explored by the group or in partnership in the future.

R1: We thank the Reviewer for the favorable evaluation. We would like to respond to these comments.

In this study, we recruited consecutive healthy patients who attended a Hungarian GP office to require the vaccinations. Thus, our aim was to conduct a “real life” study in which a typical European population was analyzed after two different types of vaccinations. Only those patients were excluded who was not COVID-19 naïve. This latter condition was not easy to set as many subjects had already been infected by the SARS-CoV-2 and we sought to investigate only the effect of homologous and heterologous vaccines via anti-SARS-CoV-2 total Ig levels. We agree with the comment that we did not measure the amount of neutralizing antibodies but based on the official description of this test (Roche Diagnostics, Mannheim, Germany), the BAU/mL values of anti-SARS-CoV-2 total Igs reflect to the level of neutralizing Igs in sera. Using this routinely available test, unfortunately, we cannot differentiate the subtypes of Igs as we measured total Ig levels.

We here focused on the humoral response induced by these different vaccine combinations, and cellular response was not studied in this study. The main reason is that whole blood samples could not be taken from these participants for flow cytometry to perform T-cell activation analysis. We are now conducting a new project to evaluate T-cell response via IFNγ levels studied by a new Roche test, but these subjects cannot be recruited for these measurements retrospectively.      

Round 2

Reviewer 1 Report

Dear authors, 

Thank you for considering my comments and suggestions and improving your work. I will now advise for its acceptance to be published in Diagnostics. 

Author Response

We are very thankful for the very positive comments. Many thanks. 

Reviewer 2 Report

The authors have addressed most of the concerns satisfactorily. I do not have further comments.

Author Response

(The authors gave the same response as above.)
